# The Spread of Peste Des Petits Ruminants Virus Lineage IV in West Africa

**DOI:** 10.3390/ani13071268

**Published:** 2023-04-06

**Authors:** Emmanuel Couacy-Hymann, Kouramoudou Berete, Theophilus Odoom, Lamouni Habibata Zerbo, Koffi Yao Mathurin, Valère Kouame Kouakou, Mohamed Idriss Doumbouya, Aminata Balde, Patrick Tetteh Ababio, Lalidia Bruno Ouoba, Dominique Guigma, Adama Dji-tombo Drobo, Mariétou Guitti, Sherry Ama Mawuko Johnson, David Livingstone Mawuko Blavo, Giovanni Cattoli, Charles E. Lamien, William G. Dundon

**Affiliations:** 1Centre National de Recherche Agronomique (CNRA), Abidjan P.O. Box 1740, Côte d’Ivoire; 2Laboratoire Central Vétérinaire de Diagnostic (LCVD) de Guinée, Conakry 001, Guinea; 3Accra Veterinary Laboratory, Veterinary Services Directorate, Accra M161, Ghana; 4Laboratoire National d’Elevage (LNE), Ouagadougou P.O. Box 907, Burkina Faso; 5Directeur National des Services Vétérinaire (DNSV), Conakry 001, Guinea; 6School of Veterinary Medicine, University of Ghana, Legon, Accra LG68, Ghana; 7Animal Production and Health Laboratory, Animal Production and Health Section, Joint FAO/IAEA Division, Department of Nuclear Sciences and Applications, International Atomic Energy Agency, P.O. Box 100, 1400 Vienna, Austria

**Keywords:** peste des petits ruminants, Burkina Faso, Côte d’Ivoire, Ghana, Guinea, phylogenetic analysis

## Abstract

**Simple Summary:**

Samples collected from goats and sheep in four West African countries (i.e., Burkina Faso, Côte d’Ivoire, Ghana and Guinea) were screened for the presence of peste des petits ruminants viruses. Positive amplicons were sequenced and shown to belong to both Lineages II and IV. This is the first time that Lineage IV has been reported in all four countries.

**Abstract:**

Monitoring the transboundary spread of peste des petits ruminants (PPR) virus is an essential part of the global efforts towards the eradication of PPR by 2030. There is growing evidence that Lineage IV is becoming the predominant viral lineage, replacing Lineage I and II in West Africa. As part of a regional investigation, samples collected in Burkina Faso, Côte d’Ivoire, Guinea and Ghana were screened for the presence of PPRV. A segment of the nucleoprotein gene from positive samples was sequenced, and phylogenetic analysis revealed the co-circulation of Lineage II and IV in Burkina Faso, Côte d’Ivoire and Guinea, and the identification of Lineage IV in Ghana. These data will be of importance to local and regional authorities involved in the management of PPRV spread.

## 1. Introduction

Given the high social and economic impact and food security implications of peste des petits ruminants (PPR), this viral disease of sheep and goats has been selected for global eradication by 2030 by the Food and Agricultural Organization (FAO) and the World Organization of Animal Health (WOAH) [1]. The disease is caused by the peste des petits ruminants virus (PPRV), a non-segmented, single-stranded, negative-sense RNA virus which is the only member of the *Small ruminant morbillivirus* species within the family *Paramyxoviridae* [2] and is characterized by high fever, ocular-nasal discharge, stomatitis, diarrhea and pneumonia. Mortality and morbidity due to PPR can be 50% to 90% and 10% to 90%, respectively, particularly in naïve populations of sheep and goats.

Although belonging to a single serotype, PPRV can be differentiated into four genetically distinct lineages at the molecular level, namely, I, II, III and IV, on the basis of based on partial sequences of the C-terminus of the nucleocapsid protein (N) gene and the fusion protein (F) gene [3,4]. All four lineages have been identified in Africa [5]. Two of the lineages (I and II) have co-circulated in West Africa for several decades following the first identification of the virus in 1942 in Côte d’Ivoire [5,6].

In order to reach the goal of globally eradicating PPR, one of the key requirements will be to have a full understanding of the viral spread and transboundary movement of infected animals in endemic countries once vaccination programs have begun. In this study, we concentrated on four countries in West Africa (i.e., Burkina Faso, Côte d’Ivoire, Ghana and Guinea) in order to characterize the PPRVs that are currently circulating in the region.

Recently, there has been strong evidence indicating the incursion of Lineage IV virus into several African countries [5,7,8,9,10]. However, up until now, there has been no published report of Lineage IV in Burkina Faso, Côte d’Ivoire, Ghana or Guinea. Lineage II has been the only lineage described in Ghana [5,11], while both Lineage I and II have been reported in Burkina Faso and Côte d’Ivoire [5,12]. More specifically, Lineage I was identified in Burkina Faso in 1988 (GenBank JN647696), while viruses characterized from outbreaks in 2007, 2008, 2009 and 2014 have been shown to belong to Lineage II [5,13]. Concerning Côte d’Ivoire, the lineage to which the PPRV identified in 1942 belonged has not been identified, but the full genome of Lineage I and Lineage II collected in 1989 and 2009, respectively, in Côte d’Ivoire are available in GenBank (EU267273 and KR781451) [12]. Of note, PPRV Lineage I viruses were present in Guinea in the late 1980s [5], indicated by the presence of two partial N and F gene sequences in GenBank (accession numbers: DQ8401770 and FR667554). Additionally, there are three GenBank submissions from 2013 of partial N gene sequences from Lineage II PPRVs from Guinea (MT072467 to MT072469).

## 2. Materials and Methods

### 2.1. Sample Description and Processing

The samples (*n* = 44) analyzed in this study were all collected following a suspicion of the disease and are described in Appendix A. The sampling locations (positive samples only) are shown in Figure 1. Briefly, 10 nasal swabs were collected in Burkina Faso between 2019 and 2021 from sheep in three different localities (Loumbila, Piela and Sabou). Twenty-one samples (organs and swabs) were collected in Côte d’Ivoire from sheep and goats between 2013 and 2021 from seven different locations (i.e., Agboville, Bocanda, Bouaké, Biankouma, Bongouanou, Grand Yapo and Dimbokro). In Guinea, 10 nasal swab samples were collected between 2021 and 2022 from six locations (Boké, Dinguiraye, Kankan, Lola, N’Zérékoré and Samoyé,) within the country, while three nasal swabs from Ghana collected in 2022 from Denyira, Navrongo and Tse Addo were provided. Total RNA purified from the respective samples using an RNeasy Mini kit (Qiagen) and standard procedures [7,9,11] was shipped to the Animal Production and Health Laboratory, Seibersdorf, Austria, for further analysis.

### 2.2. PCR Screening

A one-step RT-PCR was used to amplify a 351 bp fragment of the PPRV N gene with primers NP3 and NP4 as previously described [4].

### 2.3. Sequencing and Phlyogenetic Analysis

Positive amplicons were purified and sent for sequencing at LGC genomics (Berlin, Germany). The sequences generated have been submitted to GenBank under accession numbers OQ215758 to OQ215778, OQ448868 to OQ448873 and OQ281756. Phylogenetic trees incorporating all the sequences generated in this study combined with the others available in GenBank (the accession numbers are shown in Figure 2) were estimated using the maximum likelihood (ML) method available in MEGA 6 [14], using the Hasegawa–Kishino–Yano plus Gamma Distributed (HKY + G) model of nucleotide substitution and 1000 bootstrap replications.(Note: the analyzed N gene fragment was 219 bp, as this allowed the inclusion of all relevant sequences from GenBank). The tree was visualized and annotated using the Interactive Tree of Life (iTOL) [15].

## 3. Results and Discussion

The phylogenetic analysis of the partial N sequence identified the presence of both Lineage II and Lineage IV viruses among the samples analyzed (Figure 2). In addition, four different subclades, referred to as Subclades IV-A, IV-B, IV-C and IV-D (for the purposes of this study only) were observed (Figure 2). Note that Subclades IV-A, IV-B and IV-C were supported by bootstrap values of 90%, 94% and 78%, respectively, while Subclade IV-D was supported by a value of just 46%, indicating that care should be taken in the interpretation of this subclade.

More specifically, of the 10 samples from Burkina Faso, six (60%) were positive by RT-PCR for PPRV. Sequence and phylogenetic analysis of the amplicons revealed that these viruses belonged to Lineage II (*n* = 2) and Lineage IV (*n* = 4). The nucleotide sequences of the N gene segment of the two Lineage II viruses were identical and clustered with other West African viruses from Benin, Côte d’Ivoire (this study), Ghana, Nigeria and Mali (Figure 2). The sequences also shared 99.5% nucleotide (nt) identity with another Lineage II virus identified in Burkina Faso in 2014 [5]. (The four Lineage IV viruses were also identical to each other and to two of the Lineage IV viruses identified in Côte d’Ivoire in this study (i.e., PPRV/Goat/CôteDIvoire/BN4/2021 and PPRV/Goat/Côte DIvoire/BN6/2021) and belonged to Subclade IV-A (Figure 2).

Eighteen (78%) of the 21 samples tested from Côte d’Ivoire were positive by RT-PCR. The N gene segment sequences from 17 of these samples revealed that these viruses belonged to Lineages II (*n* = 3) and Lineage IV (*n* = 14) (Figure 2). As stated above, the Lineage II viruses were similar (99% nt identity) to those identified in Burkina Faso but were only 97.1% identical to the Lineage II viruses identified in Côte d’Ivoire in 2009. The sequences of the Lineage IV viruses from Côte d’Ivoire were heterogenous and belonged to three separate subclades (IV-A, IV-B and IV-D) (Figure 2).

Five (50%) of the 10 samples collected in Guinea were positive for PPRV. As seen in Figure 2, phylogenetic analysis revealed that four of the sequences belonged to Lineage IV and one to Lineage II. The sequence of the single Lineage II virus identified in Guinea was 97.6% similar at the nucleotide level to the viral sequence submissions of viruses from Guinea collected in 2013 (GenBank MT072467 to MT072469). The sequences of three of the Lineage IV viruses from Guinea were very similar (99.5–100% nucleotide similarity) to each other and belonged to Subclade IV-A together with viruses from Burkina Faso and Côte d’Ivoire. The similarity among the viral sequences from Guinea was not surprising, given that they were collected within three months of each other in two localities (i.e., Lola and Samoye) that are less than 50 km from each other in the southern part of Guinea close to the border with Liberia (Figure 1). In contrast, the remaining Lineage IV viral sequence, which was 5.9 to 6.0% different at the nucleotide level from the other three sequences, was from a sample collected in Kankan in the center of the country. This sequence belonged to Subclade IV-D and was very similar to the viruses from Côte d’Ivoire and Niger.

Only one (33%) of the three samples from Ghana was positive for PPRV. The sequence of this single sample also belonged to Lineage IV, clustering with sequences from Mali, Niger and Nigeria. The sequence belonged to a subclade (i.e., IV-C) (Figure 2) that was different from the subclades of the Lineage IV sequences from Burkina Faso, Côte d’Ivoire and Guinea. This indicates that the source of this Lineage IV virus may not be the same as that of the viruses identified in Burkina Faso, Côte d’Ivoire and Guinea. The collection and characterization of more samples from Ghana would clarify this observation. There is more convincing evidence from the sequence data for an epidemiological link between the Subclade IV-A viruses present in Burkina Faso, Côte d’Ivoire and Guinea and the Subclade IV-D viruses in Côte d’Ivoire and Guinea. This is supported by the fact that official trade and movement of animals between these three countries exists. In addition, pastoralist often cross borders with their flocks in search of better grasslands and water sources, while festive celebrations can also result in transboundary movements and the mingling of animals [16].

This is the first report of Lineage IV viruses in Burkina Faso, Côte d’Ivoire, Ghana and Guinea, and strongly supports the assumption that Lineage IV is not only the most widespread lineage globally but is also becoming the predominant lineage in Africa. The identification of Lineage IV viruses belonging to different subclades also corroborates data on the genetic heterogeneity of PPR viruses circulating in individual countries as reported by many authors [7,8,9,10]. In addition, our data confirmed the extensive transboundary movements of PPR within West Africa [7,9,13]. However, it is important to avoid the overinterpretation of these data, given the relatively small size (219 nt) of the N gene segment analyzed. Phylogenetic analyses based on partial N gene sequences have proven to be informative and have frequently confirmed analyses carried out using full PPRV genome sequences [5]. However, as the number of PPRVs being sequenced increases and heterogenicity is observed, particularly of Lineage IV viruses, it would be preferable to generate full genomes if more informative, in-depth analyses are to be carried out. Indeed, full genome analysis of these Lineage IV viruses together with well-designed animal studies could provide information that might explain why Lineage IV is becoming the predominant lineage in Africa.

Another limitation to this study is the small number of samples available, particularly from Ghana. Veterinary authorities should be encouraged to collect and characterize as many samples as possible before mass vaccination is implemented as part of the global eradication program [1]. This will provide important pre-vaccination data which can then be used as a baseline for post-vaccination surveillance and monitoring efforts, which will be required to confirm the eradication of PPR.

Despite these shortcomings, the data generated in this study confirm the transboundary spread of PPRV. The data should be of assistance in understanding current viral movements in the region and allow local and regional authorities to better coordinate for the eradication of PPR.

## Figures and Tables

**Figure 1 animals-13-01268-f001:**
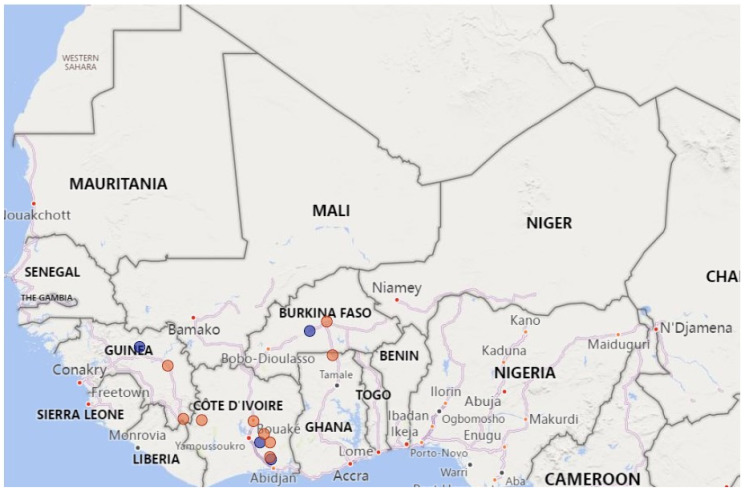
Map showing the sampling locations of PPR-positive samples. Lineage II and IV viruses are indicated by blue and red circles, respectively.

**Figure 2 animals-13-01268-f002:**
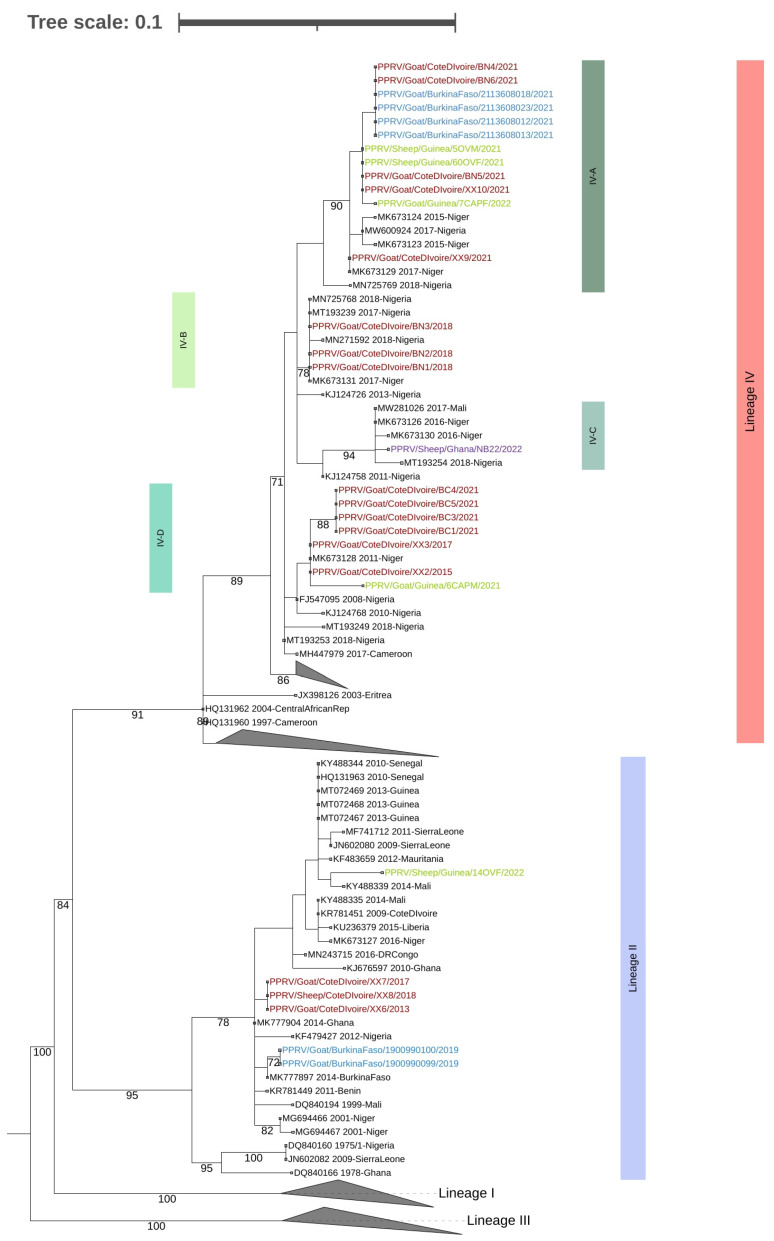
ML phylogenetic tree of the partial N gene sequences (219 nt) from the positive amplicons from this study combined with similar sequences available in GenBank. Sequences from Burkina Faso, Côte d’Ivoire, Ghana and Guinea are shown in blue, red, purple and green text, respectively. Lineages, subclades and bootstrap values > 70% are indicated. Note: For image clarity some of the sequences/subclades in Lineage IV have been collapsed. An un-collapsed tree is available as Appendix A.

## Data Availability

The sequences generated in this study have been submitted to GenBank under accession numbers OQ215762 to OQ215778, OQ448868 to OQ448873 and OQ281756.

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
