# Peer review of "The Spread of Peste Des Petits Ruminants Virus Lineage IV in West Africa"

_animals, 2023, doi:10.3390/ani13071268_

Round 1

Reviewer 1 Report

figure 1: legend says that blue dots= LIV, but I see a red dot for Ghana. Is it an issue in the legend or the figure?

 methods:

Sampling focusing on animals with PPR symptoms in all countries or only in Burkina Faso? Not clear from the text.

 more details to be provided on the type of samples (organs and swabs) and how they were prepared before RNA extraction using the Qiagen kit.

 Reduction from 351bp amplification to 219bp in the phylogenetic analysis not explained

 Results:

 Table S1

"GT" : not defined, despite asterisk

swab: type of swab?

line 19: wab written instead of swab

line 39: why N/A on this line?

 line 151-159: the logic to decide when epidemiologcial links can be made from sequence data is not well explained. Do they consider that having only one new sequence is not enough to draw conclusion? Or that you need 2-3 identical sequences from one same place to start looking at clade relationship? I would say that it depends mostly on the amount of existing additional data.

 A main issue is that they present sequences within clade IV-D as good enough for epidemiological interpretation but the clade IV-D appears to be with bootstrap support below 70%, it seems. So results should be considered with more caution than, for example, results for the Ghana sequence that is within a clade with strong bootstrap support. It is clear that you cannot say, based on these results, that all LIV PPR strains in Ghana are from a different source than strains in Burkina Faso or Guinea, but at least it is clear that this specific sequence is different than other sequences obtained in this study, so would be associated to a different chain of transmission events.

 The existence of transboundary movement of animals in the region can explain all the results obtained in terms of clades identified, without helping with pinpointing specific origin of the viruses as more detailed sampling would be needed for that. So the discussion about trade in this paragraph trade is valid for all the results obtained not only IV-A, or even less clade IV-D.

 Why mentionning this cladeIV-D with limited support, and not clade with low support in lineae II? At least highlight the fact that the bootstrap support for clade IV-D is low so interpretation should be taken with caution.

 There are many more LII sequences published from West African countries, notably from Burkina Faso and neighbouring countries. How was the dataset of published sequences selected? It would have seem logical to include all existing sequencing data from the countries that are the focus of this study. Are we sure that additional insight on relationship between the new sequences from this study and existing sequences could not have been obtained? Maybe an additional tree in supplementary material focusing on lineage II could show that nothing was missed?

 line 175: error in the sentence "why lineage IV appears are becoming"

 Missing in the discussion: the results show, as previous studies, the transboundary nature of PPRV circulation, further supporting the need for regional coordination in the effor to control PPR

Author Response

Thank you for your comments on our manuscript. A point-by-point rebuttal is provided below which we hope you find satisfactory.

Reviewer #1

figure 1: legend says that blue dots= LIV, but I see a red dot for Ghana. Is it an issue in the legend or the figure?

Response: Thank you. This error has been corrected

methods:

Sampling focusing on animals with PPR symptoms in all countries or only in Burkina Faso? Not clear from the text.

Response: the text has been revised for clarity. See Page 3 line 87

more details to be provided on the type of samples (organs and swabs) and how they were prepared before RNA extraction using the Qiagen kit.

Response: Table S1 has been updated and standard procedures have been cited. See Page 3 line 98

 Reduction from 351bp amplification to 219bp in the phylogenetic analysis not explained

Response: The reason is that some of the relevant sequences in GenBank are only 219 bp long and so in order to include them in the analysis it needed to be done with the shorter fragment.  The text has been revised to highlight this. See Page 4 line 122   

 Results:

 Table S1

"GT" : not defined, despite asterisk

Response: change to “lineage”

swab: type of swab?

Response:  information updated

line 19: wab written instead of swab m

Response: corrected

line 39: why N/A on this line?

Response: error corrected

 line 151-159: the logic to decide when epidemiologcial links can be made from sequence data is not well explained. Do they consider that having only one new sequence is not enough to draw conclusion? Or that you need 2-3 identical sequences from one same place to start looking at clade relationship? I would say that it depends mostly on the amount of existing additional data.

Response: We agree with the reviewer that our statement is too vague. We have revised the text accordingly. See page 8 lines 181-182

 A main issue is that they present sequences within clade IV-D as good enough for epidemiological interpretation but the clade IV-D appears to be with bootstrap support below 70%, it seems. So results should be considered with more caution than, for example, results for the Ghana sequence that is within a clade with strong bootstrap support. It is clear that you cannot say, based on these results, that all LIV PPR strains in Ghana are from a different source than strains in Burkina Faso or Guinea, but at least it is clear that this specific sequence is different than other sequences obtained in this study, so would be associated to a different chain of transmission events.

Response: thank you for this comment. We have revised the text to incorporate it. See page 4 lines 129-132 and page 8 lines 181-182.

The existence of transboundary movement of animals in the region can explain all the results obtained in terms of clades identified, without helping with pinpointing specific origin of the viruses as more detailed sampling would be needed for that. So the discussion about trade in this paragraph trade is valid for all the results obtained not only IV-A, or even less clade IV-D.

 Why mentionning this cladeIV-D with limited support, and not clade with low support in lineae II? At least highlight the fact that the bootstrap support for clade IV-D is low so interpretation should be taken with caution.

Response: We agree with this comment and have revised the text accordingly. See page 4 lines 129-132

There are many more LII sequences published from West African countries, notably from Burkina Faso and neighbouring countries. How was the dataset of published sequences selected? It would have seem logical to include all existing sequencing data from the countries that are the focus of this study. Are we sure that additional insight on relationship between the new sequences from this study and existing sequences could not have been obtained? Maybe an additional tree in supplementary material focusing on lineage II could show that nothing was missed?

Response: The lineage II dataset was created based on BlastN results (see  screenshots below). We believe that the current tree dataset is representative and does not need to be expanded.

 line 175: error in the sentence "why lineage IV appears are becoming"

Response: error corrected

 Missing in the discussion: the results show, as previous studies, the transboundary nature of PPRV circulation, further supporting the need for regional coordination in the effor to control PPR

Response: text revised accordingly. See page 9 lines 213-216

BlastN Screenshots

Reviewer 2 Report

Manuscript number: animals-2269101

Manuscript Title: The spread of peste des petits ruminants virus lineage IV in West Africa

Journal: Animals (MDPI)                          

The manuscript entitled “The spread of peste des petits ruminants virus lineage IV in West Africa” by Couacy-Hymann et al. is well written and the experiments are well performed, the topic and general context of the manuscript is of high interest to readers of PPR.

The manuscript should be considered as accepted for publication in Animals providing that the authors perform many amendments and language editing as follows:

Introduction:

Line 42: “Given the high social and economic impact and food security implications of Peste des Petits ruminants (PPR), …” please write “peste des petits ruminants” in small letters

Line 57: “In order to reach the goal of global eradication of PPR one of the key requirements” please add comma after PPR “In order to reach the goal of global eradication of PPR, one of the key requirements”

Line 57: “Recently there has been strong evidence” please add comma after “Recently” “Recently, there has been strong evidence”

Lines 69-71: “For Côte d’Ivoire, the lineage to which the PPRV identified in 1942 belonged has not been identified but the full genome of a lineage I and lineage II collected in 1989 and 2009 respectively in Côte d’Ivoire are available in GenBank (EU267273 and KR781451) [12]” please correct to “Concerning Côte d’Ivoire, the lineage of the PPRV identified in 1942 has not been identified but the full genome of a lineage I and lineage II collected in 1989 and 2009, respectively, in Côte d’Ivoire are available in GenBank database (accession numbers: EU267273 and KR781451) [12]”.

Lines 72-74: “According to two 72 partial N and F gene sequences in GenBank (DQ8401770 and FR667554) lineage I viruses 73 were present in Guinea in the late 1980s [5].” please correct to “Of note, PPRV lineage I viruses were present in Guinea in the late 1980s [5] indicated by presence of two partial N and F genes sequences in GenBank database (accession numbers: DQ8401770 and FR667554).”.

Lines 74-75: “Additionally, there are three GenBank submissions from 2013 of partila N gene sequences from lineage II PPRVs from Guinea (MT072467 to MT072469).” please correct to Additionally, in Guinea three GenBank submissions from 2013 belong to lineage II PPRVs (accession numbers: MT072467 to MT072469) based on the partial N gene sequences.”.

Materials and Methods:

Lines 80-82: “Briefly, ten samples were collected in Burkina Faso between 2019 and 2021 from sheep following suspicion of disease in three different localities (Loumbila, Piela, Sabou).” please correct to “Briefly, between 2019 and 2021 ten samples were collected in three different locations (Loumbila, Piela, Sabou) in Burkina Faso from sheep following suspicion of disease in three different localities (Loumbila, Piela, Sabou).”

Lines 82-84: “Twenty-one samples were collected in Côte d’Ivoire from sheep and goats between 2013 and 2021 from seven different locations (e.g. Agboville, Bocanda, Bouaké, Biankouma, Bongouanou Grand Yapo, Dimbokro).” please correct to “Between 2013 and 2021, twenty-one samples were collected from sheep and goats from seven different locations (e.g. Agboville, Bocanda, Bouaké, Biankouma, Bongouanou Grand Yapo, Dimbokro) in Côte d’Ivoire”

Lines 91-92:Figure 1. Map indicating the location of PPR postive samples. Lineage II and IV viruses are indicated by red and blue circles respectively. .” please correct to “Figure 1. Map indicating the location of PPR positive samples. Lineage II and IV viruses are indicated by red and blue circles respectively.”

Lines 94-95: “A one step RT-PCR was used to amplify a 351bp fragment of the PPRV N gene with primers NP3 and NP4 as previously described” please correct to “A one-step RT-PCR was used to amplify a 351 bp fragment of the PPRV N gene with primers NP3 and NP4 as previously described”

Results and discussion:

Lines 107-108: “The phylogenetic analysis of the partial N sequence identified the presence of both lineage II and lineage IV viruses among the sample analyzed (Figure 2).” please correct to “… the samples analyzed …”

Lines 112-113: “Sequence and phylogenetic analysis of the amplicons revealed that the viruses belonged to lineage II (n=2) and lineage IV (n=4).” please correct to “… revealed that these viruses …”

Lines 122-123: “The N gene segment sequences from 17 of these samples revealed that the viruses be-longed to lineages II (n=3) and lineage IV (n=14) (Fig 2).” please correct to “The N gene segment sequences from 17 of these samples revealed that these viruses belonged to lineages II (n=3) and lineage IV (n=14) (Figure 2).”

Lines 125-127:The sequences of the lineage IV viruses from Côte d’Ivoire were heterogenous and belonged to three sepa-rate subclades (IV-A, IV-B and IV-D) (Figure 2)” please correct to The sequences of PPRV lineage IV viruses from Côte d’Ivoire were heterogenous and belonged to three separate subclades (IV-A, IV-B and IV-D) (Figure 2).”

Figure 2.: Virus and strain names were not clear to the reader. Please enlarge the text size of Figure 2.

Lines 140-143: “The similarity among the viral sequences from Guinea was not surprising given that they were collected within three months of each other in two localities (Lola and Sam-oye) that are less than 50 kilometers from each other in the southern part of Guinea close to the border with Liberia (Fig 1).” please correct to “The similarity among the viral sequences from Guinea was not surprising given that they were collected within three months intervals from the two localities (Lola and Samoye) and the distance between localities is less than 50 kilometers where they are located in the southern part of Guinea close to the border with Liberia (Figure 1).”

Lines 143-145: “In contrast the remaining lineage IV viral sequence which was 5.9 to 6.0 % different at the nucleotide level from the other three sequences was from a sample collected in Kankan in the center of the country.” please correct to “In contrast, the remaining lineage IV viral sequence, which was different from the other three sequences at the nucleotide level (5.9 to 6.0%), was from a sample collected in Kankan in the center of the country.”

Author Response

Thank you for your comments on our manuscript. A point-by-point rebuttal is provided below which we hope you find satisfactory.

Reviewer #2

The manuscript entitled “The spread of peste des petits ruminants virus lineage IV in West Africa” by Couacy-Hymann et al. is well written and the experiments are well performed, the topic and general context of the manuscript is of high interest to readers of PPR.

The manuscript should be considered as accepted for publication in Animals providing that the authors perform many amendments and language editing as follows:

Introduction:

Line 42: “Given the high social and economic impact and food security implications of Peste des Petits ruminants (PPR), …” please write “peste des petits ruminants” in small letters

Response: corrected

Line 57: “In order to reach the goal of global eradication of PPR one of the key requirements” please add comma after PPR “In order to reach the goal of global eradication of PPR, one of the key requirements”

Response: corrected

Line 57: “Recently there has been strong evidence” please add comma after “Recently” “Recently, there has been strong evidence”

Response: corrected

Lines 69-71: “For Côte d’Ivoire, the lineage to which the PPRV identified in 1942 belonged has not been identified but the full genome of a lineage I and lineage II collected in 1989 and 2009 respectively in Côte d’Ivoire are available in GenBank (EU267273 and KR781451) [12]” please correct to “Concerning Côte d’Ivoire, the lineage of the PPRV identified in 1942 has not been identified but the full genome of a lineage I and lineage II collected in 1989 and 2009, respectively, in Côte d’Ivoire are available in GenBank database (accession numbers: EU267273 and KR781451) [12]”.

Response: revised

Lines 72-74: “According to two 72 partial N and F gene sequences in GenBank (DQ8401770 and FR667554) lineage I viruses 73 were present in Guinea in the late 1980s [5].” please correct to “Of note, PPRV lineage I viruses were present in Guinea in the late 1980s [5] indicated by presence of two partial N and F genes sequences in GenBank database (accession numbers: DQ8401770 and FR667554).”.

Response: revised

Lines 74-75: “Additionally, there are three GenBank submissions from 2013 of partila N gene sequences from lineage II PPRVs from Guinea (MT072467 to MT072469).” please correct to “Additionally, in Guinea three GenBank submissions from 2013 belong to lineage II PPRVs (accession numbers: MT072467 to MT072469) based on the partial N gene sequences.”.

Response: we do not feel that this recommended revision improves the readability of the text

Materials and Methods:

Lines 80-82: “Briefly, ten samples were collected in Burkina Faso between 2019 and 2021 from sheep following suspicion of disease in three different localities (Loumbila, Piela, Sabou).” please correct to “Briefly, between 2019 and 2021 ten samples were collected in three different locations (Loumbila, Piela, Sabou) in Burkina Faso from sheep following suspicion of disease in three different localities (Loumbila, Piela, Sabou).”

Response: the text has already been revised based on the comments of Reviewer #1 and does not require further revision

Lines 82-84: “Twenty-one samples were collected in Côte d’Ivoire from sheep and goats between 2013 and 2021 from seven different locations (e.g. Agboville, Bocanda, Bouaké, Biankouma, Bongouanou Grand Yapo, Dimbokro).” please correct to “Between 2013 and 2021, twenty-one samples were collected from sheep and goats from seven different locations (e.g. Agboville, Bocanda, Bouaké, Biankouma, Bongouanou Grand Yapo, Dimbokro) in Côte d’Ivoire”

Response: the text has already been revised based on the comments of Reviewer #1 and does not require further revision

Lines 91-92: “Figure 1. Map indicating the location of PPR postive samples. Lineage II and IV viruses are indicated by red and blue circles respectively. .” please correct to “Figure 1. Map indicating the location of PPR positive samples. Lineage II and IV viruses are indicated by red and blue circles respectively.”

Response: corrected

Lines 94-95: “A one step RT-PCR was used to amplify a 351bp fragment of the PPRV N gene with primers NP3 and NP4 as previously described” please correct to “A one-step RT-PCR was used to amplify a 351 bp fragment of the PPRV N gene with primers NP3 and NP4 as previously described”

Response: corrected

Results and discussion:

Lines 107-108: “The phylogenetic analysis of the partial N sequence identified the presence of both lineage II and lineage IV viruses among the sample analyzed (Figure 2).” please correct to “… the samples analyzed …”

Response: corrected

Lines 112-113: “Sequence and phylogenetic analysis of the amplicons revealed that the viruses belonged to lineage II (n=2) and lineage IV (n=4).” please correct to “… revealed that these viruses …”

Response: corrected

Lines 122-123: “The N gene segment sequences from 17 of these samples revealed that the viruses be-longed to lineages II (n=3) and lineage IV (n=14) (Fig 2).” please correct to “The N gene segment sequences from 17 of these samples revealed that these viruses belonged to lineages II (n=3) and lineage IV (n=14) (Figure 2).”

Response: corrected

Lines 125-127: “The sequences of the lineage IV viruses from Côte d’Ivoire were heterogenous and belonged to three sepa-rate subclades (IV-A, IV-B and IV-D) (Figure 2)” please correct to “The sequences of PPRV lineage IV viruses from Côte d’Ivoire were heterogenous and belonged to three separate subclades (IV-A, IV-B and IV-D) (Figure 2).”

Response: we do not feel that this recommended revision improves the readability of the text.

 Figure 2.: Virus and strain names were not clear to the reader. Please enlarge the text size of Figure 2.

Response: the text has been enlarged for clarity in Figure 2

Lines 140-143: “The similarity among the viral sequences from Guinea was not surprising given that they were collected within three months of each other in two localities (Lola and Sam-oye) that are less than 50 kilometers from each other in the southern part of Guinea close to the border with Liberia (Fig 1).” please correct to “The similarity among the viral sequences from Guinea was not surprising given that they were collected within three months intervals from the two localities (Lola and Samoye) and the distance between localities is less than 50 kilometers where they are located in the southern part of Guinea close to the border with Liberia (Figure 1).”

Response: we do not feel that this recommended revision improves the readability of the text.

Lines 143-145: “In contrast the remaining lineage IV viral sequence which was 5.9 to 6.0 % different at the nucleotide level from the other three sequences was from a sample collected in Kankan in the center of the country.” please correct to “In contrast, the remaining lineage IV viral sequence, which was different from the other three sequences at the nucleotide level (5.9 to 6.0%), was from a sample collected in Kankan in the center of the country.”

Response: corrected

Reviewer 3 Report

Dear Editor,

thank you for your invitation as a reviewer for the manuscript "The spread of peste des petits ruminants virus lineage IV in West Africa" in your esteemed journal "Animals". Considering the epidemiological importance of Peste des Petits Ruminants (PPR), Couacy-Hymann and coworkers investigated the spread of peste des petits ruminants virus (PPRV) lineage IV in sheep and goats in West Africa.  By one step RT-PCR,  positive amplicons were obtained and a segment of the N gene was sequenced.  Phylogenetic analysis revealed the co-circulation of lineage II and IV in Burkina Faso, Côte d’Ivoire and Guinea and the identification of lineage IV in Ghana.  Despite the relatively small size (219 nt) of the analyzed N gene segment, phylogenetic analyzes based on partial N gene sequences have frequently proven informative. The authors wants to highlight the first report of lineage IV in Burkina Faso, Côte d’Ivoire, Ghana and Guinea.

The manuscript is well-written, with appropriate methods, interpretations, and conclusions. The manuscript requires some smaller adaptations before it is suitable for publishing. Minor revision is therefore recommended.

1. Introduction

page 1 of 7:

lines 42-43- delete “Peste des Petits ruminants (PPR)” and insert “Peste des Petits Ruminants (PPR)”

page 2 of 7:

lines 47-48- the authors should check correct italics use, Small Ruminant Morbillivirus (capitalize all words) and Paramyxoviridae. Please also insert (SRMV) after Small Ruminant Morbillivirus (capitalize all words ICTV, 2019) (OIE/WOAH Terrestrial Manual 2022-CHAPTER 3.8.9. PESTE DES PETITS RUMINANTS)

line 60- replace “(e.g.” with “(i.e.” before Burkina Faso

line 64- Comment: it should be considered that in December 2022 Abel S. Biguezoton et al with a communication during the "5th Peste des Petits Ruminants Global Research and Expertise Network (PPR-GREN) Meeting" had already demonstrated the presence of PPRv lineage IV in Burkina Faso and on basis of the outbreaks investigated, they hypothesized that lineage IV replaced lineage II. Reference: 5th Peste des Petits Ruminants Global Research and Expertise Network (PPR-GREN) Meeting. Montpellier, France, 07 Dec 2022  - 09 Dec 2022. Abel S. Biguezoton, Guy Ilboudo, Barbara Wieland, Rahinata Sawadogo, Firmin Dah, Adrien Zoungrana and Michel Dione “Epidemiology of peste des petits ruminants virus in West Africa: Is lineage IV replacing lineage II in Burkina Faso?”

line 72- please, check the abnormal spacing between “in” and “genbank”

line 75- delete “partila N gene” and insert “partial N gene ”

2. Materials and methods:

2.1. Sample description and processing

page 2 of 7:

Line 79- please insert “(n=44)” after “The samples analysed” (I would also suggest to standardize the use of analyzed/analysed throughout the manuscript)

Lines 80-89- I would suggest including the type of sample carried out throughout the paragraphs. Line 80: please after “Briefly, ten samples” insert  (nasal swab)

Line 82- Comment: could you specify the type of swab (ocular, nasal…) made in Côte d’Ivoire?

Line 83- replace “(e.g.” with “(i.e.” before Agboville

Line 84- insert a comma “,” after Bongouanou

Line 85- could you specify the type of swab (ocular, nasal…) made in Guinea?; replace “(e.g.” with “(i.e.” before Boké

Lines 86-87- please after “three samples” insert  (nasal swab) from Ghana

2.3. Sequencing and phylogenetics analysis

page 3 of 7:

Line 99- please add "OQ215758-OQ215761”as reported in “Figure 2 ML phylogenetic tree…” and in "Table S1"

Line 101- delete “(accession numbers are shown in the tree figures)” and insert “(accession numbers are shown in the Figure 2)

3. Results and discussion:

page 3 of 7:

Line 108- replace “sample analyzed ” with “samples analyzed” before (Figure 2)

page 4 of 7:

Line 123- replace “(Fig 2)” with “(Figure 2)”

Line 127- insert a “period stop” after “(Figure 2)”

page 5 of 7:

Line 133- replace “Five (50%) of the ten samples” with “Five (50%) of the 10 samples” (in order to align with Burkina Faso, Côte d’Ivoire)

page 6 of 7:

Line 143- replace “(Fig 1)” with “(Figure 1)”

Line 147- replace “Only one (33%) of the three samples” with “Only one (33%) of the 3 samples” (in order to align with Burkina Faso, Côte d’Ivoire)

Lines 161-163- about Burkina faso refer to previous comment (Introduction, page 2 of 7 – line 64)

References

For all bibliographic references, choose whether or not to put a period stop “.” at the end of bibliographic citations

Line 202- Please, replace reference 1. with this corrected version: FAO OIE/WOAH Peste des Petits Ruminants. Global Eradication Programme. Contributing to Food Security, Poverty Alleviation and Resilience. Five Years 2017-2021. (2016). Available online at: http://www.fao.org/3/a-i6316e.pdf

Line 204- check the reference 2. are you sure you want to insert this reference?

Line 213- check the reference 6. please, write correctly Bull Serv Zootech Epizoot Afr Occident Fr. (1942) 5:15–21.

Line 236- check the reference 16. as follows: Apolloni, A., Corniaux, C., Coste, C., Lancelot, R., Touré, I. (2019). Livestock Mobility in West Africa and Sahel and Transboundary Animal Diseases. In: Kardjadj, M., Diallo, A., Lancelot, R. (eds) Transboundary Animal Diseases in Sahelian Africa and Connected Regions. Springer, Cham. https://doi.org/10.1007/978-3-030-25385-1_3

Figure 1

page 3 of 7:

Check the data represented in the figure and align with the legend.

Legend (Figure caption)- please, replace “Figure 1. Map indicating the location of PPR postive samples. Lineage II and IV viruses are indicated by red and blue circles respectively. .” with “Figure 1. Map showing sampling location of PPR positive samples. Lineage II and IV viruses are indicated by red and blue circles respectively.”

(Warning: in particular be careful to write "positive samples" correctly, moreover I think it is necessary to reverse the order of blue and red in the sentence in order to align the legend with the figure 1 and table S1. Remove the twice period stop at the end of the paragraph)

Figure 2

page 5 of 7:

Check the data represented in the figure and align with the legend.

Legend (Figure caption)- Warning: sequences from Burkina Faso are shown in blue, from Côte d’Ivoire in red (you need to reverse the order of blue and red in the sentence to align the legend with the figure 2)

Table S1

Supplementary file

Comment: could you specify the type of swab (ocular, nasal…) carried out  in Côte d’Ivoire and Guinea? (see also the remark in the manuscript)

Line 19 of the table (Côte d’Ivoire): please, write correctly “swab” (column sample type)

Author Response

Thank you for your comments on our manuscript. A point-by-point rebuttal is provided below which we hope you find satisfactory.

Reviewer #3

  1. Introduction

page 1 of 7:

lines 42-43- delete “Peste des Petits ruminants (PPR)” and insert “Peste des Petits Ruminants (PPR)”

Response: we do not agree with this recommendation and have revised the name as recommended by Reviewer #2 to “peste des petits ruminants (PPR)”

page 2 of 7:

lines 47-48- the authors should check correct italics use, Small Ruminant Morbillivirus (capitalize all words) and Paramyxoviridae. Please also insert (SRMV) after Small Ruminant Morbillivirus (capitalize all words ICTV, 2019) (OIE/WOAH Terrestrial Manual 2022-CHAPTER 3.8.9. PESTE DES PETITS RUMINANTS)

Response: We agree with some of the changes recommended by the reviewer. Small ruminant morbillivirus and Paramyxoviridae have been italicised. However, viral species are not abbreviated and are only capitalised at the beginning i.e. Small ruminant morbillivirus [see Kuhn, Jens H et al. “2021 Taxonomic update of phylum Negarnaviricota (Riboviria: Orthornavirae), including the large orders Bunyavirales and Mononegavirales.” Archives of Virology vol. 166,12 (2021): 3513-3566.]

line 60- replace “(e.g.” with “(i.e.” before Burkina Faso

Response: error corrected

line 64- Comment: it should be considered that in December 2022 Abel S. Biguezoton et al with a communication during the "5th Peste des Petits Ruminants Global Research and Expertise Network (PPR-GREN) Meeting" had already demonstrated the presence of PPRv lineage IV in Burkina Faso and on basis of the outbreaks investigated, they hypothesized that lineage IV replaced lineage II. Reference: 5th Peste des Petits Ruminants Global Research and Expertise Network (PPR-GREN) Meeting. Montpellier, France, 07 Dec 2022  - 09 Dec 2022. Abel S. Biguezoton, Guy Ilboudo, Barbara Wieland, Rahinata Sawadogo, Firmin Dah, Adrien Zoungrana and Michel Dione “Epidemiology of peste des petits ruminants virus in West Africa: Is lineage IV replacing lineage II in Burkina Faso?”

Response: Thank you for this information. Unfortunately, this communication does not appear in a peer-reviewed publication.

line 72- please, check the abnormal spacing between “in” and “genbank”

Response: corrected

line 75- delete “partila N gene” and insert “partial N gene ”

Response: corrected

  1. Materials and methods:

2.1. Sample description and processing

page 2 of 7:

Line 79- please insert “(n=44)” after “The samples analysed” (I would also suggest to standardize the use of analyzed/analysed throughout the manuscript)

Response: text revised as recommended

Lines 80-89- I would suggest including the type of sample carried out throughout the paragraphs. Line 80: please after “Briefly, ten samples” insert  (nasal swab)

Response: text revised as recommended

Line 82- Comment: could you specify the type of swab (ocular, nasal…) made in Côte d’Ivoire?

Response: information provided

Line 83- replace “(e.g.” with “(i.e.” before Agboville

Response: error corrected

Line 84- insert a comma “,” after Bongouanou

Response: error corrected

Line 85- could you specify the type of swab (ocular, nasal…) made in Guinea?; replace “(e.g.” with “(i.e.” before Boké

Response:  information provided

Lines 86-87- please after “three samples” insert  (nasal swab) from Ghana

Response: the text has been revised based on the comment

2.3. Sequencing and phylogenetics analysis

page 3 of 7:

Line 99- please add "OQ215758-OQ215761”as reported in “Figure 2 ML phylogenetic tree…” and in "Table S1"

Response: Thank you for highlighting this omission. The text has been revised accordingly.

Line 101- delete “(accession numbers are shown in the tree figures)” and insert “(accession numbers are shown in the Figure 2)

Response: revised as recommended

  1. Results and discussion:

page 3 of 7:

Line 108- replace “sample analyzed ” with “samples analyzed” before (Figure 2)

Response: corrected

page 4 of 7:

Line 123- replace “(Fig 2)” with “(Figure 2)”

Response: corrected

Line 127- insert a “period stop” after “(Figure 2)”

Response: corrected

page 5 of 7:

Line 133- replace “Five (50%) of the ten samples” with “Five (50%) of the 10 samples” (in order to align with Burkina Faso, Côte d’Ivoire)

Response: corrected

page 6 of 7:

Line 143- replace “(Fig 1)” with “(Figure 1)”

Response: corrected

Line 147- replace “Only one (33%) of the three samples” with “Only one (33%) of the 3 samples” (in order to align with Burkina Faso, Côte d’Ivoire)

Response: corrected

Lines 161-163- about Burkina faso refer to previous comment (Introduction, page 2 of 7 – line 64)

Response: see previous response

References

For all bibliographic references, choose whether or not to put a period stop “.” at the end of bibliographic citations

Response: corrected

Line 202- Please, replace reference 1. with this corrected version: FAO OIE/WOAH Peste des Petits Ruminants. Global Eradication Programme. Contributing to Food Security, Poverty Alleviation and Resilience. Five Years 2017-2021. (2016). Available online at: http://www.fao.org/3/a-i6316e.pdf

Response: corrected

Line 204- check the reference 2. are you sure you want to insert this reference?

Response: yes, this reference is suitable

Line 213- check the reference 6. please, write correctly Bull Serv Zootech Epizoot Afr Occident Fr. (1942) 5:15–21.

Line 236- check the reference 16. as follows: Apolloni, A., Corniaux, C., Coste, C., Lancelot, R., Touré, I. (2019). Livestock Mobility in West Africa and Sahel and Transboundary Animal Diseases. In: Kardjadj, M., Diallo, A., Lancelot, R. (eds) Transboundary Animal Diseases in Sahelian Africa and Connected Regions. Springer, Cham. https://doi.org/10.1007/978-3-030-25385-1_3

Response: corrected

Figure 1

page 3 of 7:

Check the data represented in the figure and align with the legend.

Legend (Figure caption)- please, replace “Figure 1. Map indicating the location of PPR postive samples. Lineage II and IV viruses are indicated by red and blue circles respectively. .” with “Figure 1. Map showing sampling location of PPR positive samples. Lineage II and IV viruses are indicated by red and blue circles respectively.”

Response: corrected

(Warning: in particular be careful to write "positive samples" correctly, moreover I think it is necessary to reverse the order of blue and red in the sentence in order to align the legend with the figure 1 and table S1. Remove the twice period stop at the end of the paragraph)

Response: corrected

Figure 2

page 5 of 7:

Check the data represented in the figure and align with the legend.

Legend (Figure caption)- Warning: sequences from Burkina Faso are shown in blue, from Côte d’Ivoire in red (you need to reverse the order of blue and red in the sentence to align the legend with the figure 2)

Response: corrected

Table S1

Supplementary file

Comment: could you specify the type of swab (ocular, nasal…) carried out in Côte d’Ivoire and Guinea? (see also the remark in the manuscript)

Response: text revised as recommended

Line 19 of the table (Côte d’Ivoire): please, write correctly “swab” (column sample type)

Response: corrected

Reviewer 4 Report

The authors in their short communication stated that this study was undertaken with an aim to fully understand the viral spread and transboundary movement of infected animals, one of the key requirements in endemic countries before vaccination programs have begun to  reach the goal of global eradication of PPR and

the authors claimed this is the first report of lineage IV viruses in the studied countries and becoming the predominant lineage in west Africa.

The authors are well-renowned international PPR experts and this study is supported by the fact that official trade and movement of animals between these three countries exist in Africa.

Minor comments

Abbreviation for APHL should be expanded in the text

Author Response

Thank you for your comments on our manuscript. A point-by-point rebuttal is provided below which we hope you find satisfactory.

Reviewer #4

Abbreviation for APHL should be expanded in the text

Response: APHL expanded as recommended